# Anti-Inflammatory Effects of Allocryptopine via the Target on the CX3CL1–CX3CR1 axis/GNB5/AKT/NF-κB/Apoptosis in Dextran Sulfate-Induced Mice

**DOI:** 10.3390/biomedicines11020464

**Published:** 2023-02-05

**Authors:** Yang Yang, Tingyu Ding, Gang Xiao, Jialu Huang, Dan Luo, Meishan Yue, Yue Su, Sujuan Jiang, Jianguo Zeng, Yisong Liu

**Affiliations:** 1College of Veterinary Medicine, Hunan Agricultural University, Changsha 410125, China; 2Loudi Vocational and Technical College, Loudi 417000, China; 3College of Veterinary Medicine, Shanxi Agricultural University, Taiyuan 030031, China

**Keywords:** allocryptopine, inflammatory bowel disease, chemokine signaling pathway, apoptosis, CX3CL1–CX3CR1 axis, central nervous system

## Abstract

Allocryptopine (ALL) is an isoquinoline alkaloid extracted from *Macleaya cordata*
**(Willd). R. Br.**, which has been claimed to have anti-inflammatory and neuroprotection properties. However, the mechanism by which ALL ameliorates inflammatory bowel disease (IBD) remains unclear. Here, we used network pharmacology and quantitative proteomic approaches to investigate the effect of ALL on IBD pathogenesis. Network pharmacology predicted potential targets and signaling pathways of ALL’s anti-IBD effects. As predicted by network pharmacology, gene ontology (GO) analysis, in terms of the proteomic results, showed that the immune response in mucosa and antimicrobial humoral response were enriched. Further study revealed that the ALL-related pathways were the chemokine signaling pathway and apoptosis in the Kyoto Encyclopedia of Genes and Genomes (KEGG). In addition, we identified AKT1 as a hub for the critical pathways through protein–protein interaction (PPI) network analysis. Similar to mesalazine (MES), Western blot verified that ALL downregulated upstream chemokine CX3CL1 and GNB5 content to reduce phosphorylation of AKT and NF-κB, as well as the degree of apoptosis, to improve inflammatory response in the colon. Our research may shed light on the mechanism by which ALL inhibits the CX3CL1/GNB5/AKT2/NF-κB/apoptosis pathway and improves the intestinal barrier to reduce colitis response and act on the CX3CL1–CX3CR1 axis to achieve neuroprotection.

## 1. Introduction

Inflammatory bowel disease (IBD) mainly consists of ulcerative colitis (UC) and Crohn’s disease (CD), and its pathogenic mechanism remains incompletely defined. In the current review, we demonstrated that IBD incidence and progression can be synergistically attributed to mucosal barrier degradation, disordered gut microbiota, and intestinal immune dysregulation [1]. Mesalazine, 5-Aminosalicylic acid, is now frequently used to treat IBD [2]. Recently, more novel biologic treatments including cytokines inhibitors, Janus kinase inhibitors, and antitrafficking therapies have been developed for multifactorial-induced IBD. Targeted inflammatory signaling pathways are currently being inhibited as part of therapeutic approaches [3].

The chemokine signaling pathway is gradually demonstrating its importance in the onset and progression of multiple diseases. Chemokines were discovered for their initial role in the immune system and inflammation response; subsequent studies have since shown that the chemokines also are involved in guiding the migration of neurons and cancer cells [4]. There are many structures of chemokines, such as CC, CXC, CX3C, and XC [5]. As the sole member of the chemokine CX3C family, CX3CL1 is unique and binds to its corresponding specific receptor, CX3CR1. Once CX3CL1 conjugates with CX3CR1, the CX3CL1/CX3CR1 axis can initiate the activation of downstream inflammatory signaling pathways such as the PI3K-AKT, MAPK, and NF-κB pathways, demonstrating bidirectional communication between the CX3CL1–CX3CR1 axis and the NF-κB pathway. The CX3CL1/CX3CR1 axis can raise the NF-κB phosphorylation level through the control of Iκ-Bα, which, in turn, facilitates CX3CL1–CX3CR1 binding. In IBD, the involvement of the CX3CL1–CX3CR1 axis regulates macrophage and intestinal immune function, and deletion of CX3CL1 and CX3CR1 results in the translocation of the gut microbiome and enhances IBD severity [6]. The CX3CL1–CX3CR1 axis controls various signaling pathways, and as a result, it plays a role in a number of clinical diseases such as neuropathy. CX3CL1 and CX3CR1 localize to neurons and microglia, respectively, leading to communication between the two cell types under abnormal conditions [7]. Studies have found that inhibition of CX3CL1 signaling can reduce neurotoxicity and neuroinflammation in rats. In contrast, this finding also showed reduced CX3CL1 levels in the Alzheimer’s disease autopsy brain sections. Whereas the effects of CX3CR1 deficiency are neurotoxic toward the late stages of Aβ pathology, they may be protective in the early stages [8]. Therefore, we can easily conceive that the CX3CL1–CX3CR1 axis has a dynamic background-dependent effect on neuropathy. Taken together, the above evidence highlights the importance of the CX3CL1–CX3CR1 axis and the chemokine signaling pathway in multiple diseases, especially neurological disorders and IBD.

Allocryptopine (ALL) is an isoquinoline alkaloid from plants of the Papaveraceae family. These plants, which to the Macleaya herb, have been utilized in Europe, America, and Asia for a long time due to their anti-inflammatory and antibacterial effects [9]. The authors of a previous report speculated that ALL may be metabolized in the gut and undergo enterohepatic circulation in rats [10]. Recent research has demonstrated that ALL plays a neuroprotective role by suppressing oxidative-stress-induced neuronal apoptosis [11]. Additionally, an increasing number of studies have shown the antiarrhythmic effects of ALL [12,13], although no study has been conducted regarding the potential of these alkaloid extracts as alternative drug candidates in colitis. In our study, we investigated the protective effect of ALL on DSS-induced IBD through a change in colon length and confirmed that ALL can reverse colon shortening in DSS-induced mice. Subsequent research investigated the mechanism of ALL in DSS-induced colitis using network pharmacology and label-free quantitative proteomics. Together with bioinformatics analysis and Western blot verification, we found that ALL reduced the accumulation of upstream chemokine CX3CL1 to inhibit NF-κB and apoptosis pathways to achieve anti-inflammatory action and regulated tight-junction proteins occludin and claudin-1 to repair the intestinal barrier in the colon. It was reported that neuroinflammation plays a significant role in the development and progression of neurodegenerative disease and CX3CL1–CX3CR1 signaling targeted for the treatment of neurodegeneration [14]. Neurodegenerative diseases (ND) such as AD and Parkinson’s disease (PD) were enriched in the KEGG according to network pharmacology and proteomics, so we inferred that ALL may act on the CX3CL-CX3CR1 axis and suppress neuroinflammation to achieve neuroprotection.

## 2. Materials and Methods

### 2.1. GEO Data Acquisition and Identification of DEGs

We downloaded the gene expression profiles of GSE22307, GSE34874, GSE42768, and GSE67577 from the GEO datasets and chose samples of the control and IBD model mice. These datasets meet the following criteria: all IBD model mice were induced by DSS. We applied GEO2R, an online analysis tool, to analyze the degree of data discrepancy between the control and the DSS groups in GEO to screen the DEGs. In addition, false discovery rate (FDR) < 0.05 and |logFC| > 1 were set as the criteria to screen out DEGs.

### 2.2. Network Pharmacology and Molecular Docking

Potential targets of allocryptopine were predicted using SwissTargetPrediction (http://www.swisstargetprediction.ch/). Targets for IBD are available in the GeneCards database (https://www.genecards.org/), DisGeNET (https://www.disgenet.org/), and OMIM (https://www.omim.org/). Gene symbols were obtained and checked in the UniProt database (http://www.UniProt.org/) for the species condition Homo sapiens. The structures of the target proteins and compounds were downloaded from the RCSB PDB (https://www.rcsb.org) and PubChem (https://pubchem.ncbi.nlm.nih.gov/) databases, respectively. The storage format of the compound structure was transformed by Openbabel 2.4.1 software. Solvent and organic molecules in the protein receptors were removed using PyMOL 2.2 software. Molecular docking was performed with AutoDock 4.2.6 software. The AutoGrid 4 and AutoDock 4 modules were used to enable semiflexible docking and obtain affinities for small-molecule compounds and protein receptors.

### 2.3. Animals

Sixty male C57BL/6 mice of SPF class weighing 20 ± 2 g (license number SCXK (Xiang) 2019–0014) were purchased from Tianqin Biotechnology Co. (Changsha, China). They were kept at a temperature of 20 to 24 °C, a relative humidity of 50% to 70%, and alternating day and night light for 12 h. They were free to drink and feed.

### 2.4. Establishment of a 3% DSS-Induced IBD Model in Mice Treated with Mesalamine and Allocryptopine

A total of 40 mice were allowed to adapt for 7 days and were then assigned randomly into four groups (n = 10 per group), including a control group (control), DSS-induced colitis model group (DSS), DSS-induced colitis mice treated with a dose of mesalamine (200 mg/kg) group (DSS + MES), and a DSS-induced colitis mice treated with an amount of Allocryptopine (50 mg/kg) group (DSS + ALL). Allocryptopine (98%), C_21_H_23_NO_5_, MW 369.41 was purchased from Hunan Meikeda Biological Resource Co. (Changsha, China), (batch number 170601). DSS (M.W.: 36,000–50,000) was obtained from MP Biomedicals (Santa Ana, CA). Mesalamine was purchased from Meilunbio Co. (Dalian, China). After adaptation, the control group and the DSS group were given 1% Carboxymethylcellulose sodium (CMC-Na, CAS: 9004-32-4) by oral administration. The mice in the other groups received a daily oral injection of each test compound for seven days and were permitted to drink and eat normally during that period. On the eighth day, the control group continued to drink regular water with intragastric administration of 1% CMC-Na. In comparison, the other groups were induced by adding 3% DSS (*w*/*v*) to their drinking water. Furthermore, the DSS group continued to be given 1% CMC-Na, and corresponding drugs were applied to the treatment groups within the next seven days. After euthanasia by CO_2_, colons were collected and rinsed with precooled saline, blotted with clean filter paper, rapidly snap-frozen in liquid nitrogen, and stored at −80 °C.

### 2.5. Label-Free Quantitative Proteomics

#### 2.5.1. Protein Sample Preparation

Protein was extracted from tissue samples using 100 µL SDT lysis buffer. The samples were boiled for 3 min and further ultrasonicated for 2 min at 4 °C. Undissolved cellular debris was removed by centrifugation at 16,000× *g* for 20 min. The supernatant was collected and quantified with a BCA protein assay kit.

#### 2.5.2. Protein Digestion

Protein digestion (300 μg for each sample) was performed with filter-aided sample preparation (FASP) by adding 1 M DTT to a final concentration of 100 mM, boiling for 5 min, and cooling to room temperature. Then, 200 µL of UA buffer (8 M Urea in 0.1 M Tris-HCl, pH 8.5) was added to 1 mL of sample, mix welled, transferred to a 10 KD ultrafiltration tube, and centrifuged at 12,000× *g* for 15 min. Subsequently, 200 µL of UA buffer was added and centrifuged at 12,000× *g* for 15 min, followed by the addition to 100 µL of iodoacetamide (IAA: 0.05 M Iodoacetamide in UA) and 0.1 mL/sample, shaken at 600 rpm for 1 min, kept away from light for 30 min at room temperature, and centrifuged at 12,000× *g* for 10 min. Then, 100 µL of UA buffer was added, centrifuged at 12,000× *g* for 10 min, and repeated twice. Subsequently, 100 µL NH_4_HCO_3_ buffer was added, centrifuged at 14,000× *g* for 10 min, and repeated twice, followed by the addition of 40 µL trypsin buffer (6 µg Trypsin in 40 µL NH_4_HCO_3_ buffer) and shaken at 600 rpm for 1min at 37 °C for 16–18 h. The collection tube was replaced with a new one and centrifuged at 12,000× *g* for 10 min; then, the filtrate was collected, the peptides that were desalted using a C18 cartridge were added and lyophilized under vacuum. The enzymatically digested peptides were then dried in 0.1% TFA. The peptide concentration was determined for LC-MS analysis.

#### 2.5.3. LC-MS/MS Analysis

LC-MS/MS was performed on a Q Exactive Plus mass spectrometer coupled with an Easy 1200 nLC (Thermo Fisher Scientific, Bremen, Germany). The peptide was first loaded to a trap column (100 μm × 20 mm, 5 μm, C18, Dr. Maisch GmbH, Ammerbuch, Germany) in buffer A (0.1% Formic acid in water). Reverse-phase high-performance liquid chromatography (RP-HPLC) separation was performed with an EASY-nLC system (Thermo Fisher Scientific, Bremen, Germany) using a self-packed column (75 μm × 150 mm; 3 μm ReproSil-Pur C18 beads, 120 Å, Dr. Maisch GmbH, Ammerbuch, Germany) at a flow rate of 300 nL/min. The RP−HPLC mobile phase A was 0.1% formic acid in water, and B was 0.1% formic acid in 95% acetonitrile. Peptides were eluted over 120 min with a linear gradient of buffer B. MS data were acquired using a data-dependent top20 method, dynamically choosing the most abundant precursor ions from the survey scan (300–1800 *m*/*z*) for HCD fragmentation. The instrument was run with peptide recognition mode enabled. A lock mass of 445.120025 Da was used as an internal standard for mass calibration. The full MS scans were acquired at a resolution of 70,000 at *m*/*z* 200 and 17,500 at *m*/*z* 200 for MS/MS scan. The maximum injection time was set to 50 ms for MS and 50 ms for MS/MS. The normalized collision energy was 27, and the isolation window was set to 1.6 Th. The dynamic exclusion duration was 60 s.

#### 2.5.4. Sequence Database Searching and Data Analysis

The MS data were analyzed using Proteome Discoverer software (version 2.4, Thermo Scientific). MS data were searched against the UniProt-Mus musculus (mouse) (88,108 total entries, downloaded 13 September 2021). Trypsin was selected as the digestion enzyme. The maximal two missed cleavage sites and one peptide tolerance of 10 ppm were defined for database search. Carbamidomethylation of cysteine was specified as a fixed modification, whereas acetylation of the protein N terminal, oxidation of methionine, and deamidation of N and Q were specified as variable modifications for database searching. The database search results were filtered and exported with <1% FDR at the peptide-spectrum-matched level and protein level, respectively. Label-free quantification was carried out in Proteome Discoverer using an intensity determination and normalization algorithm as previously described. The “LFQ intensity” of each protein in different samples was calculated as the best estimate, satisfying all pairwise peptide comparisons. This LFQ intensity was almost on the same scale as the summed-up peptide intensities. The quantitative protein ratios were weighted and normalized by the median balance in Proteome Discoverer software. Only proteins with fold change ≥1.5 and a *p*-value < 0.05 were considered for significantly differential expressions.

#### 2.5.5. Bioinformatics Analysis

Analyses of bioinformatics data were performed with Perseus software, Microsoft Excel, and R statistical computing software. Hierarchical clustering analysis was performed with the heatmap package based on the open-source statistical language R25, using Euclidean distance as the distance metric and the complete method as the agglomeration method. Information was extracted from UniProtKB/Swiss-Prot, KEGG, and GO to annotate the sequences. GO and KEGG enrichment analyses were carried out with Fisher’s exact test, and FDR correction for multiple testing was also performed. GO terms were grouped into three categories: biological process (BP), molecular function (MF), and cellular component (CC). Enriched GO and KEGG pathways were nominally statistically significant at the *p* < 0.05 level.

### 2.6. Western Blot Analysis

Total tissue lysates were separated on SDS-PAGE and electrotransferred onto polyvinylidene fluoride membranes (0.22 μm) by wet blotting (Bio-Rad). Primary antibodies detected the following proteins: β-actin (1:100,000, Cat No. AC026, Abclonal), α-tubulin (1:50,000, Cat No. 66031-1-Ig, Proteintech), CX3CL1 (1:1000, Cat No. A14198, Abclonal), GNB5 (1:1000, Cat No. A4447, Abclonal), AKT2 (1:5000, Cat No. ab131168, Abcam), p-AKT (1:1000, Cat No. YP0006, Immunoway), NF-κB (1:2000, Cat No. 66535-1-Ig, Proteintech), p-NF-κB (1:1000, Cat No. 93H1, Cell Signaling Technology), Bad (1:1000, Cat No. ab32445, Abcam), Bcl-2 (1:2000, Cat No. ab182858, Abcam), occludin (1:3500, Cat No. 13409-1-AP, Proteintech), claudin-1 (1:4500, Cat No. 13050-1-AP, Proteintech), which were then incubated with goat anti-rabbit IgG and goat anti-mouse IgG and HRP-conjugated(1:5000, Cat No. CW0103S, CW0102S, CWBIO). Then, stripping buffer (CW0056M, CWBIO) was used for the replacement of antibodies. The density of each protein was detected by an ECL detection kit (Solarbio).

### 2.7. PPI Network Analysis and Identification

Protein–protein interaction (PPI) networks were constructed using the STRING database (https://string-db.org), and we defined a confidence score > 0.4 as the cutoff criterion for the interaction between the two proteins. Cytoscape software was applied for visualization of the topological network and the PPI network.

### 2.8. Statistical Analyses

All the experiments were tested three times. All data were expressed as the mean plus SD. For regular data analysis, the differences among groups were assessed by one-way analysis of variance (ANOVA) followed by Tukey’s post hoc multiple comparisons test. Statistical analyses were performed using GraphPad Prism 8.0.2 (GraphPad, San Diego, CA, USA). Multiple comparisons an one-way ANOVA from GraphPad Prism 8.0.2 software were used to compare the mean of each column with the mean of every other column under Tukey’s multiple comparisons test. Statistical significance was determined by one-way ANOVA with Tukey’s test for multigroup comparisons. We used the ‘Wu Kong’ platform (https://www.omicsolution.com/wkomics/main/) assessed on 13 September 2021 to analyze the network pharmacology of GO and KEGG enrichment, comparable GEO data, hierarchical cluster analysis (HCA), and GSEA KEGG enrichment [15].

## 3. Results

### 3.1. Network Pharmacology and Bioinformatics Analysis

A total of 100 targets were obtained by predicting ALL using SwissTargetPrediction, and a total of 1578, 6442, and 186 IBD-associated targets were separately obtained from the DisGeNET, GeneCards, and OMIM public databases, respectively (Figure 1A). We further identified the common 68 ALL–IBD intersecting targets as key genes for subsequent bioinformatics analysis. GO enrichment showed the top 20 entries in the three classifications under ALL-related bioenrichment, including biological processes such as protein phosphorylation, apoptotic process, and innate immune response. The subcellular location of proteins is mainly in the cellular environment, such as the plasma membrane and nucleus. Molecular functions include ATP binding and protein serine/threonine kinase activity (Figure 1B). The PI3K-AKT signaling pathway, apoptosis, Alzheimer’s disease, and chemokine signaling pathway were all among the top 50 signaling pathways, as shown in Figure 1C. Then, the intersecting targets were imported into the STRING database for protein–protein interaction analysis, and the results were imported into Cytoscape 3.7.2 software to map the PPI network. The top seven network targets for mTOR, SRC, MAPK3, PIK3CA, PI3KCB, MAPK1, and SIRT1 are displayed in Figure 1D.

### 3.2. Inflammatory Indicators Were Increased in the DSS-Induced Model

GSE22307, GSE34874, GSE42768, and GSE67577 are sample gene expression microarrays with both regular and DSS-induced colon samples. A bar plot figure was generated to visualize the significantly differentially expressed inflammatory indicators based on GEO2R analysis (Figure 2). Increased mRNA levels of CC and CXC chemokine subfamilies, interleukins, and their receptors were observed in the DSS-induced mice. It is evident that DSS treatment alters the gene expression of chemokines, interleukin, and their related factors, indicating IBD. Thus, the chemokine signaling pathway sheds light on a novel target for ALL in treating IBD.

### 3.3. Molecular Dock of ALL to the Four Core Targets

Figure 3 shows the molecular docking of ALL to the key proteins (CX3CL1, CX3CR1, AKT, and NF-κB p65) in the chemokine signaling pathway. The results reveal that ALL had good binding characteristics to the four key targets and that the alkaloid may act on these four targeted proteins. In turn, the chemokine signaling pathway, as well as the PI3K-AKT and NF-κB pathways, was impacted.

### 3.4. Identification of DEPs between Treatments

Following 3% DSS uptake, IBD manifestations developed in mice, as evidenced by the colon becoming red and shorter with no formed content. Treatment of ALL and MES showed the therapeutic effects on DSS-induced murine IBD, as evidenced by reversed colon shortening and the formation of contents in the intestine (Figure 4A and Appendix A). This implies that ALL had the same ability to treat IBD as MES.

To elucidate the therapeutic mechanism of MES and ALL in DSS-induced mice, LC-MS/MS profiling was performed on colon samples. First, 7095 protein groups were detected in colon samples. Secondly, PCA analysis was carried out to visualize the general differences among samples. The PCA scatter plot shows that samples from the ALL+DSS group and the MES+DSS group were closer to the control group than the DSS group at the PC1 axis and the PC2 axis, respectively (Figure 4B). A comparison between the DSS group and the control group revealed a total of 784 DEPs. Furthermore, 383 DEPs and 372 DEPs were identified in the MES and ALL treatment groups compared to the DSS group. These results imply that the protein level in the colon was considerably impacted by both MES and ALL treatments. For better differentiation, upregulated or downregulated DEPs are indicated by red and blue bars, respectively. The bar chart shows that MES and ALL drug applications increased by 157 DEPs and 179 DEPs. In contrast, the DSS group was reduced by 226 DEPs and 193 DEPs, respectively (Figure 4C). In the MES and ALL treatment groups, both the number of both down- and upregulated proteins was lower than in the DSS group, indicating that the protein expression affected by DSS was minimized or compensated by MES and ALL pretreatment, which suggests a potential anti-inflammatory effect of ALL on the colon similar to MES.

### 3.5. Gene Ontology (GO) Enrichment Analysis between Treatments

To understand the functions of DEPs, a gene ontology enrichment analysis was conducted by mapping to nodes in the gene ontology database and categorized as BP, CC, or MF (Figure 5). DEPs performed a variety of functions between the DSS and ALL+DSS groups, such as innate immune response in the mucosa, antimicrobial humoral response, and leukocyte migration involved in inflammatory response under the category of BP; mRNA editing complex, presynaptic endosome membrane, and high-density lipoprotein particles under the category of CC; and Toll-like receptor 4 binding, cytochrome-c oxidase activity, and peptidoglycan binding under the category of MF. Overall, these GO terms indicate the significant impact of ALL on anti-inflammation, antimicrobiology, and immune response.

### 3.6. Kyoto Encyclopedia of Genes and Genomes (KEGG) Metabolic Pathway Enrichment Analysis between Treatments

Comparison of the DSS group to the control group based on the rich factor showed that the upregulated pathways, which include the complement and coagulation cascades, SNARE interactions in vesicular transport, Staphylococcus aureus infection, fat digestion and absorption, and nitrogen metabolism, were all enriched (Figure 6A). The downregulated metabolic pathways in the MES treatment group included DNA replication, steroid biosynthesis, mismatch repair, biosynthesis of unsaturated fatty acids, and SNARE interactions in vesicular transport (Figure 6B). ALL was also negatively regulated for porphyrin and chlorophyll metabolism, sulfur metabolism, nitrogen metabolism, drug metabolism–cytochrome P450, and tryptophan metabolism (Figure 6C).

Significantly different expressions of apoptosis, tryptophan metabolism, the Toll-like receptor signaling pathway, Alzheimer’s disease, and the chemokine signaling pathway were presented in different groups (Figure 6D), confirming that both MES and ALL improved colitis via several pathways. A previous report that ALL can inhibit acetylcholinesterase (AChE) led us to assess the impact of ALL on AD [16]. Another study found that ALL works through antiapoptosis and anti-oxidative stress to achieve anti-Alzheimer’s action [11]. Furthermore, the CX3CL1–CX3CR1 axis connects the chemokine signaling pathway and Alzheimer’s disease. Additionally, the inflammatory response is inextricably linked to apoptosis and oxidative stress. Thus, we hypothesize that ALL improves colitis related to Alzheimer’s disease, apoptosis, and the chemokine signaling pathway.

### 3.7. Protein Changes Related to Key KEGG Pathways

The crucial pathways related to the chemokine signaling pathway and apoptosis were enriched by KEGG analysis. To further understand the impact of ALL, MES, and DSS on the key pathways, we used HCA to explore the correlation of the related protein molecules.

CX3CL1, an upstream protein of the chemokine pathway, was upregulated after DSS stimulation alone and returned to normal after MES and ALL. Moreover, CX3CL1 is commonly present throughout neural cells and mediates microglial migration. In the heat map of a chemokine signaling pathway (Figure 7A), ALL is shown to reduce the protein levels of Mapk3, Prkacb, Crk, Rock2, Csk, Grb2, Gnb5, Rac1, Gnb1, Rela, and Chuk, which were increased by DSS; however, ALL upregulated Akt1, Stat3, Rock1, Dock2, Gnb2, and Stat1. These data imply that the chemokine signaling pathway interacted with inflammatory signaling pathways such as PI3K-AKT, Jak-Stat, and NF-κB. This conclusion is consistent with the previous observation that ALL affected CX3CL1 content to mediate signaling pathways related to the subsequent downstream inflammatory signaling pathways.

Analysis of DEPs related to the inflammatory signaling pathway in the KEGG database indicates that apoptosis is one of the subsequent downstream signaling pathways of the chemokine signaling pathway (Figure 7B). According to the cluster analysis heat map of apoptosis, the protein expression in the ALL treatment group was similar to that of the control group, which was distinct from the DSS stimulation, such as caspase 3, caspase 7, Prkar1a, Tradd, Cycs, Bcl2, Chuk, Akt2, Rela, etc. Interestingly, Akt1 and Akt2 from the DSS and ALL groups presented opposite trends in the apoptosis signaling pathway, and the Akt2 condition from the ALL group presented a similar trend to the control, unlike the DSS group. As in previous studies [17], the knockdown of Akt1 enhanced Akt2 content and apoptotic resistance. This implies that colitis might be associated with the interaction of Akt1 and Akt2, which are important kinases mediating DSS-induced apoptosis.

Gene set enrichment analysis (GSEA) of the KEGG plot showed that the corresponding proteins of the chemokine signaling pathway were predominantly upregulated by ALL applications in DSS-induced mice. ALL resulted in both up- and downregulation of various proteins in the apoptosis pathway (Figure 7C). As shown in network figures, these proteins are involved in the chemokine signaling pathway and apoptosis, which further demonstrates a linkage between ALL and multisignaling pathways.

### 3.8. Western Blot Analysis of Related Protein Levels

To verify the accuracy of the proteomic data under different treatments, Western blot analysis was performed to reflect the complexity of related proteins. Based on the Western blot (Figure 8), the high expression of CX3CL1 presented in DSS-induced mice was greatly reduced by ALL and MES treatment. When CX3CL1 binds to its receptor, CX3CR1, the signal is transmitted to Gnai, which leads to its separation from Gnb5. Gnai then promotes the PI3K-AKT and NF-κB signaling pathways and, in turn, causes abnormal apoptosis, as shown by increased protein content of the Gnb5, Akt2, p-AKT, NF-κB p65, p-p65, and Bcl-2, as well as decreased Bad in the DSS stimulation group. The anti-inflammatory ability of ALL could suppress the abovementioned protein changes closer to the normal level, which has been attributed to the direct regulation of CXs3CL1 by ALL. In summary, ALL inhibits monocyte aggregation by the CX3CL1–CX3CR1 axis to control the inflammatory response. In addition, ALL could ameliorate the barrier damage in colitis, as indicated by changes in the ALL group in the levels of occludin and claudin-1 converging to the control group.

### 3.9. Protein–Protein Interaction (PPI) Network Analysis of Related Proteins of Key KEGG Pathways

PPI is the basis for displaying biological functions in the chemokine signaling pathway and apoptosis. However, further analysis of the PPI network showed that these two pathways also involve protein interaction. In terms of key proteins linked with these pathways (Figure 9A), the network revealed a strong relationship between proteins, containing 110 target nodes and 1967 edges. Further analysis showed that Akt1 and Mapk3 had the highest degree among the 110 related proteins, which were engaged in the tight-junction, Toll-like receptor signaling pathway and the MAPK signaling pathway, except for two focused pathways. In the conjoint analysis between PPI (Figure 9) and HCA (Figure 7), we speculated that Akt1 participated in signal transmission to control inflammation response via interaction with Akt2. Further cluster analysis showed that Grb2, Cdc42, Ptk2, and Ptk2b had the highest degree (Figure 9B). Akt1, which is related to these multiple signaling pathways, may be a hub contributing to improved DSS-induced colitis after ALL treatment (Figure 9C).

## 4. Discussions

Previous studies have indicated that ALL acts as a potential neuroprotective agent due to its antioxidative and antiapoptotic capabilities [11]. Moreover, it has been claimed that ALL can affect the signal transduction ability of the GABA _A_ receptor [18]. Dysfunction of GABA led to an imbalance between excitatory and inhibitory activity, which promoted the spread of Aβ and Tau pathology, and was considered to be a potential factor for AD [19]. However, no relevant research has been reported on the anti-IBD effect for ALL. Attributed to anti-inflammation ability, more natural alkaloids extracted from the planta were found to be effective in IBD therapy [20,21,22]. Berberine (an isoquinoline alkaloid) was reported to mediate the interaction between neuroimmune activity and IBD by acting on adhesive molecules and chemokines [23]. Additionally, Chen et al. reported that berberine regulated gut microbiota and fecal metabolites to ameliorate IBD. Moreover, Ge et al. confirmed that berberine is involved in tryptophan metabolites to ameliorate depressive symptoms [24]. The abovementioned studies remind us that ALL may also avoid aggregation of adhesive molecules and chemokines to manage IBD and indirectly affect neuroinflammation via the brain–gut axis. Via the application of network pharmacology and label-free quantitative proteomics, our study was designed to elucidate how ALL reverses IBD and to determine whether the anti-IBD effect of ALL is associated with neuroimmune pathways.

According to GO enrichment analysis (Figure 5), the current data confirmed that ALL mainly affects innate immune response in the mucosa, as well as antimicrobial humoral response and leukocyte migration involved in inflammatory response in BP. These results further support established a of link between ALL, the gut microbiota, and immune and inflammatory responses. Notably, ALL and MES downregulated the enrichment of high-density lipoprotein (HDL) particles in CC. It has been widely proven that HDL levels reflect the immune function of the organism and that HDL can bind to LPS and LTA, the generated Gram-negative and Gram-positive bacteria, respectively, which is attributed to its anti-inflammatory and antibacterial properties [25]. Consistent with HDL, the defense response to Gram-positive and Gram-negative bacteria is enriched in BP. According to KEGG enrichment analysis (Figure 6), tryptophan metabolism was largely enriched in the ALL group and acted as a prominently downregulated enrichment pathway in ALL+DSS vs. the DSS group. Previous research pointed out that tryptophan catabolites activate AhR, the regulator of intestinal immunity, based on gut microbiota, suggesting that ALL might regulate the gut microbiota to maintain immune function and resist IBD [26,27]. Furthermore, Wu et al. proposed that the majority of the tryptophan-transformed pathway is the kynurenine pathway (KP) and that tryptophan–kynurenine metabolism is a link between the gut for IBD and the brain for depression. Indoleamine 2,3-dioxygenase, a rate-limiting enzyme in the KP, was mainly expressed on astrocytes and microglia in the intracerebral environment, which is the key to communicating the CNS and tryptophan metabolism [28]. This indicates that ALL may regulate IBD and the central nervous system through tryptophan metabolism based on the immune system and intestinal flora.

Western Blot verified that the level of CX3CL1 was significantly lower in the MES and ALL groups than the control group (Figure 8). CX3CL1, a unique chemokine, is abundantly expressed in neurons and binds to CX3CR1 to activate signaling molecules, including AKT, IKK, and NF-κB. CX3CL1 was recently observed in the colon of IBD patients and Oxa-treated mice, whereas the use of anti-CX3CL1 mAb can significantly improve colitis symptoms by dislodging the crawling monocytes and repressing leukocyte activation and infiltration [29]. In addition, CX3CL1 is primarily located in microglia cells within the central nervous system (CNS). The CX3CL1–CX3CR1 axis involves a neuroimmune pathway via the communication between neurons and microglia [7]. In the case of inflammation, CX3CL1 is highly expressed by epithelial intestinal cells, and its targeted cells, the microglia cells, are most closely related to the mononuclear phagocytic system both ontogenetically and functionally [30]. It appears that CX3CL1 regulates the neuroimmune system to affect inflammatory response.

Moreover, Gampierakis et al. reported that both acute and chronic colitis influences the nervous and innate immune systems [31], which is in accord with IBD patients’ higher frequency of cognitive impairment and psychiatric disorders [32]. Patients with IBD also have an increased risk of developing neurodegenerative diseases (NDs), such as PD [33,34,35]. There is an inextricable link between IBD and neurodegenerative diseases. IBD is life-long disease, the pathology of which primarily affects the intestine, although it has far-reaching effects beyond the gut [36]. Talley et al. confirmed that DSS-treated mice exhibited not only the clinical features of colitis but also neuroinflammation, as evidenced by increased inflammatory cytokines, as well as caspase and microglia activation in the brain [33,37]. This inflammatory phenotype is similar to the signature detected in LPS-induced mice. Substantial evidence supports the contribution of neuroinflammation to ND, including AD and PD [38,39]. In patients with NDs such as AD and PD, IL-1β, IL-6, TNF-α, and TGF-β were elevated [38,40,41,42,43]. Therefore, we believe that inflammation is the connection between IBD and ND. Understanding the mechanism of this connection between IBD and neuroinflammation is necessary to develop natural multifunctional anti-inflammatory natural products to prevent IBD and IBD-induced CNS dysfunction.

The significant signaling pathways in the immune system activated in IBD and neuroinflammation are the NF-κB transcription factor pathway and apoptosis. The translocation of NF-κB could cause apoptosis and eventually lead to the production and release of proinflammatory cytokines and increased inflammation [44,45,46]. Dapagliflozin has been reported to exhibit marked antineuroinflammation effects in a murine model of PD by inhibiting neuronal oxidative stress and apoptosis and curbing the activation of the NF-κB pathway [47]. Furthermore, Hany et al. reported that dapagliflozin also mitigated colitis severity through suppression of oxidative stress, NF-κB signaling, and apoptosis in the colon [48]. This shows that neuroinflammation and intestinal inflammation are possible links through oxidative stress and inflammatory pathways such as apoptosis and NF-κB. When anti-inflammatory compounds exert anti-inflammatory effects at multiple sites in an organism, both neuroinflammation and intestinal inflammation may be alleviated. This condition is also seen in anti-inflammatory probiotics. Kim et al. verified that anti-inflammatory probiotics, including NK151, NK173, and NK175, alleviated neuroinflammation and colitis by the suppression of the IL-1β or IL-6–10 expression ratio [49].

Furthermore, the gut microbiota was reported to control the maturity and function of microglia in the CNS via fermentation products, i.e., SCFAs [50]. Circulating levels of the SCFAs acetate and valerate, as well as proinflammatory cytokines, were positively correlated with Aβ content in the brains of AD and PD patients [51], inferring that neuropathy was accompanied by a shift in the inflammation level and SCFA metabolism. Federica et al. demonstrated that the gut microbiota influences microglial functions via the CX3CL1–CX3CR1 axis [52]. These findings support the hypothesis that host microbiota-regulated microglia progress through bidirectional communication in the CX3CL1–CX3CR1 axis may be the key link between the gut and the CNS, and our study revealed that ALL has an effect on the CX3CL1–CX3CR1 axis and antibacterial ability. Therefore, we inferred that ALL neuroprotection depends on the abovementioned effects.

Numerous native ingredients used in traditional Chinese medicine have been reported to exhibit anti-IBD effects and interactions with other diseases, such as cancer, neurodegenerative diseases, etc. [53]. Taken together, ALL has the potential for the treatment of IBD and to provide neuroprotection. One explanation for this phenomenon is that it has anti-inflammatory and antiapoptotic abilities. Furthermore, ALL acts on the CX3CL1–CX3CR1 axis, which may be the key to the brain–gut axis. However, the present study is subject to limitations, including the absence of analysis of the gut flora and tryptophan metabolism and detection of related proteins in the cerebral cortex and hippocampus of IBD mice.

## 5. Conclusions

Network pharmacology predicted the targets of ALL in IBD; the results indicate that ALL has the potential to treat IBD. According to label-free quantitative proteomics, the colon proteome changed in the ALL group and the DSS group. Further analysis and Western blot validation identified the CX3CL1/GNB5/AKT2/NF-κB/apoptosis pathway as the ALL treatment mechanism in the DSS-induced IBD model, suggesting the neuroprotection effects of ALL through crosstalk of the CX3CL1–CX3CR1 axis and the neuroimmune system.

## Figures and Tables

**Figure 1 biomedicines-11-00464-f001:**
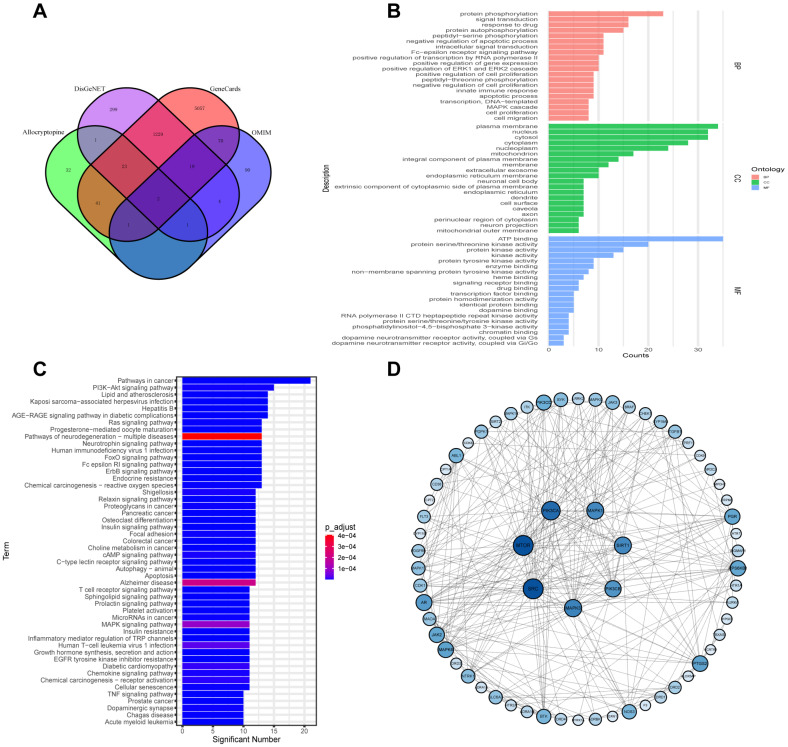
Network pharmacology and bioinformatics analysis for ALL–IBD intersecting targets. (**A**) VENN diagram of the number of potential targets of ALL vs. IBD. (**B**,**C**) GO enrichment and KEGG analysis of the 68 ALL–IBD intersecting targets. (**D**) The PPI network of the 68 ALL–IBD intersecting targets was constructed using Cytoscape software.

**Figure 2 biomedicines-11-00464-f002:**
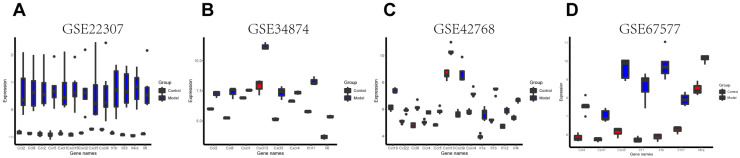
Inflammatory indicators were increased in DSS-induced models. (**A**–**D**) Gene expression of inflammatory indicators was analyzed in four microarray databases: GSE22307, GSE34874, GSE42768, and GSE67577.

**Figure 3 biomedicines-11-00464-f003:**
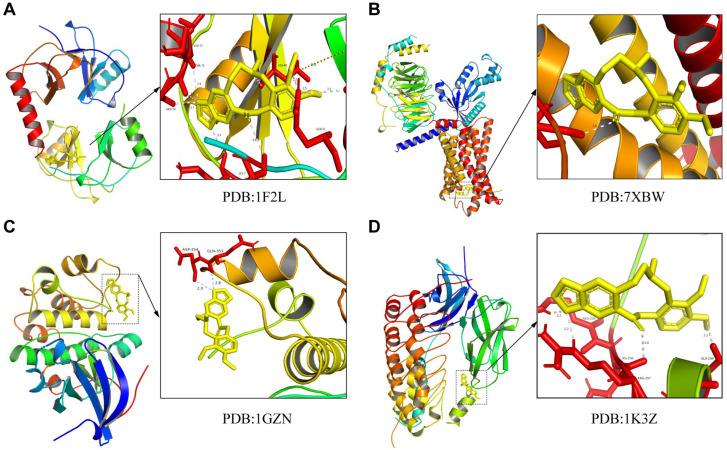
Molecular dock of ALL to CX3CL1 (**A**), CX3CR1 (**B**), AKT (**C**), and NF-κB p65 (**D**).

**Figure 4 biomedicines-11-00464-f004:**
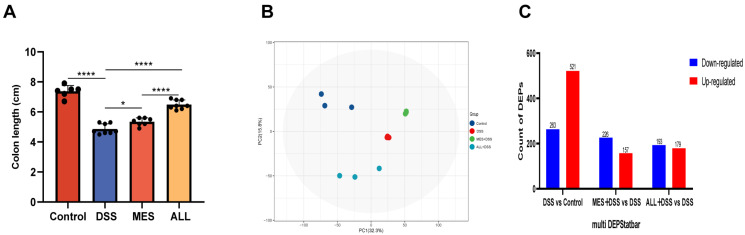
The effects of difference treatments. (**A**) ALL improved the colon length of DSS-induced colitis in C57BL/6 mice. * *p* < 0.05 and **** *p* < 0.0001. (**B**) The PCA score was different in each group. (**C**) Up- and downregulated DEPs of each comparison group. Each point in the graph represents a sample, and the distance between the points reflects the sample similarity. Only proteins with a fold change ≥1.5 and a *p*-value < 0.05 were considered for DEPs.

**Figure 5 biomedicines-11-00464-f005:**
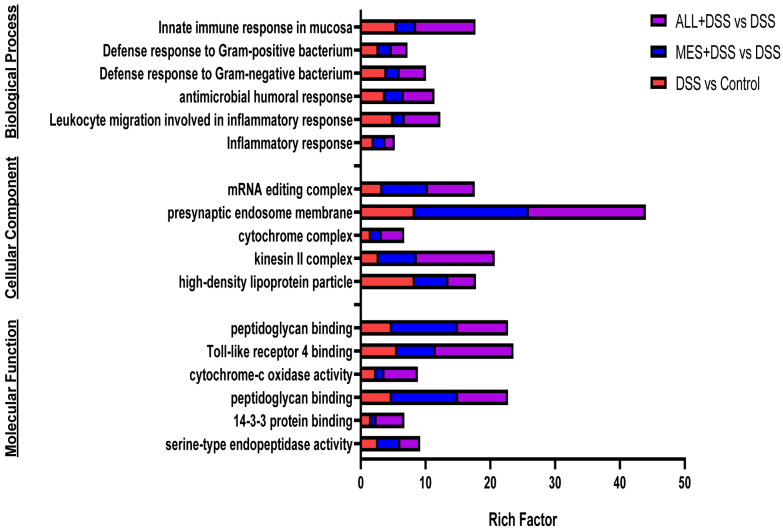
GO enrichment analysis of differentially expressed proteins (DEPs). The y-axis represents GO terms, including BP, CC, and MF, and the x-axis shows the rich factor in different treatments compared with the DSS.

**Figure 6 biomedicines-11-00464-f006:**
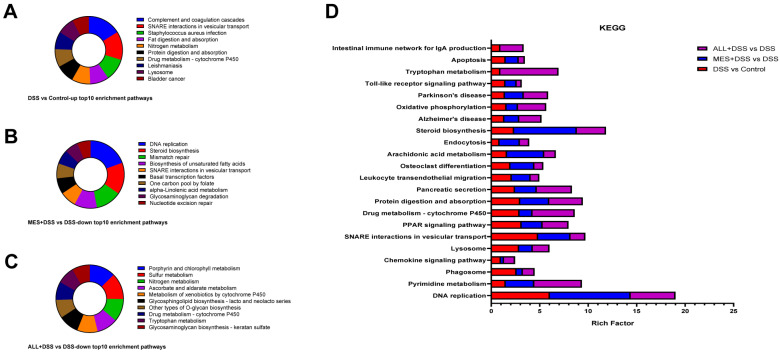
Distribution of prominent enrichment pathways (left) and enrichment analysis of differentially expressed proteins (DEPs) in the KEGG metabolic pathway (right). (**A**–**C**) Pie charts showing the top 10 enrichment pathways in different treatments compared with DSS. (**D**) The y-axis represents pathway terms, and the x-axis corresponds to the rich factor.

**Figure 7 biomedicines-11-00464-f007:**
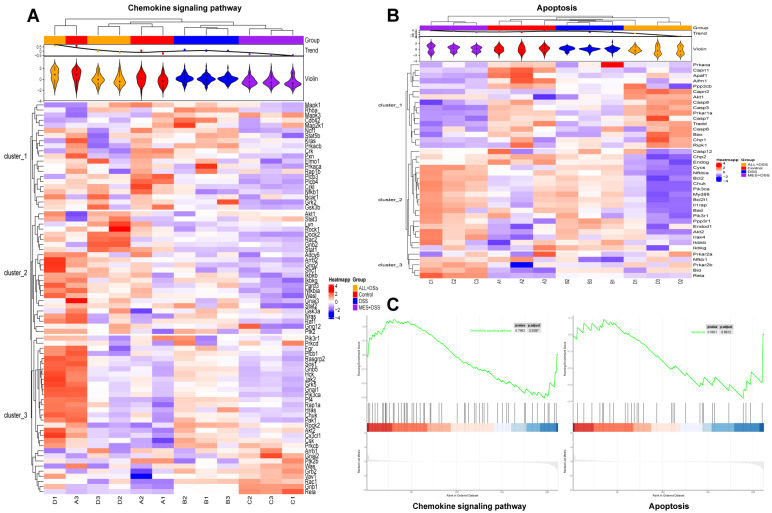
Protein changes in the chemokine signaling pathway and apoptosis. Hierarchical cluster analysis of differentially expressed proteins related to the chemokine signaling pathway (**A**) and apoptosis (**B**). Each group has three repetitions, and the violin plot indicates uniformity of biological repetitions. The color blocks at different positions represent the relative expression level of protein at the corresponding position; red represents high expression, and blue represents low expression. (**C**) GSEA KEGG enrichment analysis for the chemokine signaling pathway and apoptosis pathways.

**Figure 8 biomedicines-11-00464-f008:**
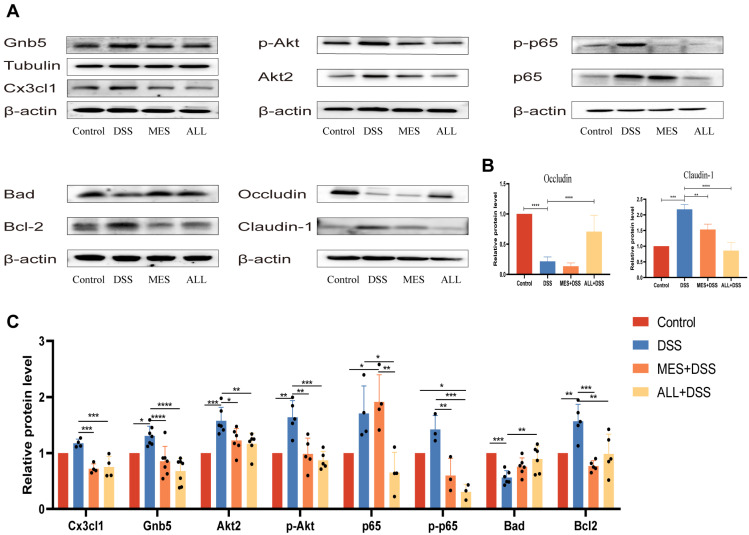
Effects of ALL and MES in DSS-induced mice on protein levels of Cx3cl1, Gnb5, Akt2, p-Akt, p65, p-p65, Bad, Bcl-2, occludin, and claudin-1. Proteins were determined by Western blot, grayscale analysis was performed using Image J, and relative protein content was calculated; the measured data were expressed as mean ± SD (ratio to β-actin/Tubulin) and analyzed by one-way ANOVA. * *p* < 0.05, ** *p* < 0.01, *** *p* < 0.001, and **** *p* < 0.0001. All trials were repeated at least three times.

**Figure 9 biomedicines-11-00464-f009:**
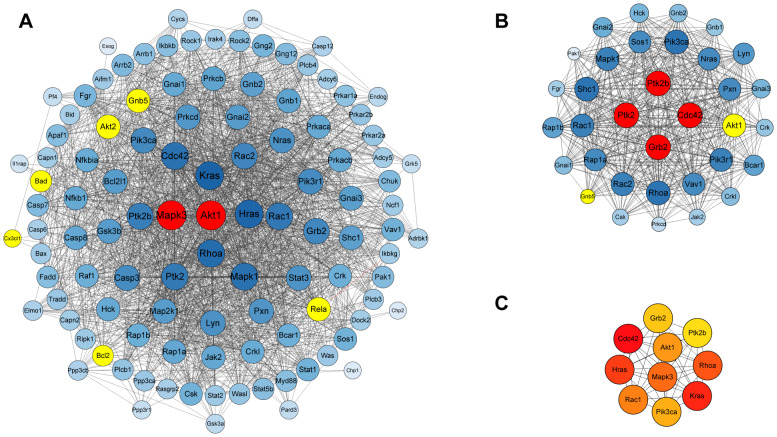
The PPI networks of proteins related to the chemokine signaling pathway and apoptosis. (**A**) PPI networks of all proteins from the two key pathways. (**B**) The cluster was separated by MCODE, and the score was 28.788. The size and color gradient of nodes indicate the degree of the proteins. The red core nodes represent the highest degree. The proteins verified by Western blot analysis are highlighted in yellow. (**C**) The network was obtained with the top 10 Hubba nodes ranked by MCC using cytoHubba. The darker the red color, the more vital the proteins.

## Data Availability

The data presented in this study are available upon request from the corresponding author. The mass spectrometry proteomics data were deposited with the ProteomeXchange Consortium via the PRIDE partner repository with the dataset identifier PXD038736.

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
