# Peer review of "Anti-Inflammatory Effects of Allocryptopine via the Target on the CX3CL1–CX3CR1 axis/GNB5/AKT/NF-κB/Apoptosis in Dextran Sulfate-Induced Mice"

_biomedicines, 2023, doi:10.3390/biomedicines11020464_

Round 1

Reviewer 1 Report

This article is generally interesting and my comments aim to increase the scientific soundness and clarity of it. English grammar and syntax in the manuscripts must be checked and corrected by a native English-speaking person. 

Line 76-78 – Please explain properly your hypothesis.

Line 98-100 – Please provide details of software mentioned: Openbabel, PyMOL, AutoDock and AutoGrid. Also licenses if needed should be provided.

Line 104 – Please provide Ethic Committee agreement for all procedures involving animals.

Line 113 – Details of Mesalamine and Allocryptopine producer is missing

Line 115 – what CMC-Na stands for ?

Line 122 – From histological point of view there are only four kinds of tissues: epithelial, connective, muscular and nervous. Therefore, such terms as “colonic tissue” (line 122, 268) is not justified. The authors terribly confuse organs with tissues.

Line 130 – please explain FASP, IIA, UA

Line 138 – please change to NH4HCO3

Line 192 – details of primary and secondary antibodies are missing (codes, concentrations, references etc.)

Line 208 – what was post-hoc test in ANOVA?

Line 285 – BP, MF and CC were already abbreviated in lines 189-190.

Figure 1, 9 – are too small to read.

Line 309 – the term “expression” should be restricted to genes only.

Figure 8 – please provide full blots images. 

Author Response

Dear reviewer:

Thank you for your letter with the comments concerning our manuscript entitled " The proteomic and network pharmacology approaches to reveal the protective effect of Allocryptopine via the target on CX3CL1-CX3CR1 axis/GNB5/AKT/NF-κB/Apoptosis in DSS-induced IBD". Those comments are all valuable and very helpful for revising and improving our manuscript, as well as the important guiding significance to our researchers. We have studied the comments carefully and have made corrections which we hope meet with approval. Revised portions are marked in yellow in the manuscript. The main responses to the comments are as follow (Line numbers correspond to simple markers of the revision pattern).

Reviewer 2 Report

In this study, the authors used pharmacological approaches and proteomic analysis of DSS treated mice +/- Allocryptopine to study the protective effect of Allocryptopine on IBD mice model (DSS). The authors reported that CX3CL1-CX3CR1 axis could be the protective pathway in IBD DSS model or Alzheimer disease.

Major comments

1) It is not convincing that this pathway CX3CL1-CX3CR1  could be a protective pathway for Alzheimer disease.  Especially the samples analyzed were colon, not brain. Also, the DSS model is not a suitable model for studying Alzheimer disease.

2) The authors did not provide any data about the protective effect of Allocryptopine such as colon length, histology of colon, disease index,  inflammatory transcripts  ...etc

3) All western blots in the supplementary are not original uncropped gel

4) Results and Discussion should be changed not to include any data about Alzheimer disease. 

Author Response

(The authors gave the same response as above.)

Round 2

Reviewer 1 Report

The authors reasonably answered my concerns.

One more issue needs to be clarified:

1. Please specify what method was used to electrotransfer proteins to membranes (wet, semi-dry or dry)?

Author Response

(The authors gave the same response as above.)

Reviewer 2 Report

The authors replied to my comments, but they do not upload the uncropped gel.

Author Response

Dear reviewer:

Thank you for your letter with the comments concerning our manuscript entitled " The proteomic and network pharmacology approaches to reveal the protective effect of Allocryptopine via the target on CX3CL1-CX3CR1 axis/GNB5/AKT/NF-κB/Apoptosis in DSS-induced IBD". Those comments are all valuable and very helpful for revising and improving our manuscript, as well as the important guiding significance to our researchers. We have studied the comments carefully and have made corrections which we hope meet with approval. Revised portions are marked in yellow in the manuscript. The main responses to the comments are as follow.

Round 3

Reviewer 2 Report

NA

Author Response

Thank you for your helpful comments on our work. According to your suggestions, we have supplemented several data in the revised marked-up manuscript and corrected several mistakes in our previous draft. Based on your comments, we also attached a point-by-point response to you. We feel great thanks for your professional review work on our study. Best wishes!